# Physiological and Transcriptomic Responses of Growth in *Neolamarckia cadamba* Stimulated by Exogenous Gibberellins

**DOI:** 10.3390/ijms231911842

**Published:** 2022-10-06

**Authors:** Lu Li, Jiaqi Wang, Jiajun Chen, Zhihua Wang, Mirza Faisal Qaseem, Huiling Li, Aimin Wu

**Affiliations:** 1State Key Laboratory for Conservation and Utilization of Subtropical Agro-Bioresources, South China Agricultural University, Guangzhou 510642, China; 2Guangdong Key Laboratory for Innovative Development and Utilization of Forest Plant Germplasm, College of Forestry and Landscape Architectures, South China Agricultural University, Guangzhou 510642, China

**Keywords:** GA, *Neolamarckia cadamba*, physiology, transcriptome analysis

## Abstract

(1) The phytohormones gibberellins (GAs) play a crucial role in plant growth and development, such as seed germination, flowering, fruiting, and stem elongation. Although many biological roles of GAs have been studied intensively, the molecular mechanisms of GAs in woody plants are still unclear. (2) In this study, we investigated the effects of exogenous application of GAs on *Neolamarckia cadamba*. (3) The height and biomass of *N. cadamba* increased after 7 days of GA treatment, especially on the second internode. Transcriptome analysis showed that although the majority of genes involved in the GA signaling pathway were up-regulated, the expression of GA20 oxidase (GA20ox) and GA3 oxidase (GA3ox) was down-regulated in the 3 days GA-treated group compared to the CK group. The expression of the cell elongation-related basic helix-loop-helix genes *bHLH74* and *bHLH49* was up-regulated in the GA-treated group compared with the CK group. Transcriptional expression levels of transcription factors involved in hormone signaling were changed, mainly including bHLH, ethylene response factor (ERF), and WRKY families. In addition, the transcriptional expression level of the key enzymes engaged in the phenylalanine pathway was downregulated after GA treatment. (4) In brief, our findings reveal the physiological and molecular mechanisms of exogenous GA treatment stimulation in *N. cadamba*.

## 1. Introduction

Phytohormones, including auxin, ethylene, cytokinin, gibberellins (GAs), abscisic acid (ABA), and brassinosteroids (BRs), participate in various biological processes during plant development and growth [1,2]. GAs play an important part in the crop green revolution by being involved in seed germination, flowering, fruit formation, stem elongation, and other developmental activities [3,4]. The physiological functions of GAs in plant development and growth have been widely studied. According to the research results in Arabidopsis, GAs signaling regulates secondary cell wall formation, particularly xylem expansion and xylem fibers synthesis in hypocotyls, offering a fresh perspective for wood improvement [5,6,7]. In *Populus*, bioactive GAs signaling is detected in developing xylem and the key enzyme GA20ox promoted cell proliferation [8] and xylem width [6]. Furthermore, GAs were found to be involved in plant photosynthesis regulation by influencing the expression of photosynthesis genes [9,10,11].

GAs are primarily generated in plant stem, root, flower, and fruit, and the terpenoid pathway is involved in their synthesis process [12,13,14]. Enzymes ENT-COPALYL PYROPHOSPHATE SYNTHASE (CPS), ENT-KAURENE SYNTHASE (KS), ENT-KAURENE OXIDASE (KO), and ENT-KAURENOIC ACID OXIDASE (KAO) catalyze the conversion of GERANYLGERANYL DIPHOSPHATE (GGDP) to GA_12_, and finally GA_12_ is converted to different forms of GAs catalyzed by GA2-oxidase (GA2ox), GA3-oxidase (GA3ox) and GA20-oxidase (GA20ox) [15,16,17]. Only a few of the several types of GAs produced in plants, primarily GA1, GA3, GA4, and GA7, are bioactive regulators that contribute to plant development and growth via the GAs signal transduction system [18,19]. Several components of the GAs signaling system, including positive and negative regulators, have been investigated [20,21].

In the presence of bioactive GAs, proteolytic degradation of DELLA proteins, which function as a master regulator component and are essential to the GAs signaling pathway, resulting in an active GA-regulated response [22,23,24]. Bioactive GAs bind to the receptor protein GIBBERELLIN INSENSITIVE DWARF1 (GID1) in rice and enhance the interaction between the DELLA protein, resulting in the degradation of DELLA through polyubiquitylation. The SCF^SLY1/GID2^ ubiquitin E3 ligase complex is required for polyubiquitination and subsequent degradation by the 26 S proteasome to regulate leaf and stem elongation [25,26,27].

Exogenous GA_3_ treatment in particular enhances xylem development to increase shoot and root length and alters the expression level of GAs biosynthesis and signaling pathway genes, as well as other hormones and cell wall associated genes in plants [28], indicating that GAs play an important role in transcription regulation and crosstalk with other phytohormone signaling pathways during plant development regulation.

*N. cadamba*, a fast-growing member of the *Rubiaceae* family species, can grow into tall trees in a short time and have been widely planted and utilized in south Asia [29,30]. Previous research has found that various nutrient levels such as potassium (K), aluminum (Al), nitrogen (N), and boron (B) have a role in the regulation of *N. cadamba* growth [31,32,33,34]. Here, the response of *N. cadamba* to exogenous GAs is studied at the physiological and molecular levels.

We reported that exogenous GAs promote the second internode development by regulating cell elongation in *N. cadamba*. RNA-seq data from different GA-treated stages analysis indicated that both the GA synthesis pathway and GA signaling pathway are affected. Furthermore, GA treatment altered the expression levels of genes involved in the phenylalanine pathway, as well as phytohormones and transcription factors Altogether, we found that exogenous GAs do not alter plant growth and development but do affect the GA pathway and other phytohormone responses in *N. cadamba*.

## 2. Results

### 2.1. Effect of Exogenous GA Application on N. cadamba Growth

To investigate the effect of exogenous GA on *N. cadamba* growth, we performed pilot experiments. Five different concentration gradients levels (0, 25, 50, 75, 100 mg/L) of GA were tested and 50 mg/L GA was selected as the suitable concentration [11]. Plants cultivated in a liquid nutrient solution containing exogenous 50 mg/L GA were referred as the GA group, and they were compared to plants grown under normal conditions (no exogenous GA, CK group). Although there was no significant difference in plant height between the GA and CK groups after 3 d, the plant height, particularly the second internode, was higher in the GA group at 7 d and 14 d (Figure 1A,B). The plant length and second internode expansion rate in the 7 d GA group were 18.4% and 74.1%, respectively, which were higher than the CK group. Furthermore, the plant length and second internode expansion rate in the 14 d GA group were 33.7% and 91.4%, respectively, when compared to the CK group (Figure 1C). Furthermore, at 7 days, the shoot fresh weight average increase rates of the CK and GA groups were 21.1% and 53.6%, respectively, and at 14 days, the shoot fresh weight average increase rates of the CK and GA groups were 53.4% and 105.1%, respectively. At 7 days, the shot dry weight average rise rates for the CK and GA groups were 29.4% and 55.0%, respectively, and at 14 days, the shoot dry weight average increase rates for the CK and GA groups were 157.8% and 210.1%, respectively (Figure 1D). Meanwhile, we measured the fresh and dried weights of the roots and found no significant difference between GA and CK. We concluded, tentatively, that exogenous GA increased the development of *N. Cadamba*, particularly at the second internode.

### 2.2. Identification of DEGs in N. cadamba in GA Treatment

Based on the physiological changes in 1 d to 14 d *N. cadamba* plants of the GA group, we further investigated the underlying molecular mechanism of GA effect on *N. cadamba* growth. The total RNA was extracted from the second internode of the CK group and GA group at 1, 3, 7, and 14 days and named as CK1d, CK3d, CK7d, CK14d, GA1d, GA3d, GA7d, and GA14d, respectively. After the removal of low quality and short reads, clean reads mapped over 75% of the *N. cadamba* reference genome (Appendix A). Principal component analysis (PCA) for all samples revealed that samples from each-group were clustered together, indicating good correlation among samples (Figure 2A). Twelve commonly differentially expressed genes (DEGs) of the second internode were identified in four treatment time points by using a Venn diagram (Figure 2B). There were 1021, 518, 947, and 3146 DEGs in response to GA treatment and CK at 1 d, 3 d, 7 d, and 14 d, respectively (Figure 2C).

### 2.3. Expression Profiles of DEGs

To further investigate the effects of exogenous GA on the molecular mechanism of *N. cadamba* plant growth, the DEGs were analyzed with GO, KEGG and KOG databases.

DEGs were classified into three main GO categories, namely biological process (BP), cellular component (CC), and molecular function (MF). Different pathways, including defense response (GO:0006952), DNA replication (GO:0006260), homeostatic process (GO: 0042592) and cellular homeostasis (GO:0019725) are enriched in the BP category. Thylakoid membrane (GO:0042651) and extracellular region (GO:0005576) pathways were enriched in the CC category. Additionally, hydrolase activity, acting on tetrapyrrole binding (GO:0046906); glycosyl bonds (GO:0016798); heme binding (GO:0020037); catalytic activity, and acting on DNA (GO:0140097) pathways were enriched in the MF category (Appendix A).

Moreover, DEGs enriched in KEGG database pathways were found to be significantly up- or down-regulated by GA treatment. The DEGs identified in CK1d vs. GA1d were mainly enriched in the carbon metabolism pathway, and the DEGs from GA3d vs. CK3d, GA7d vs. CK7d, and GA14d vs. CK14d were mainly enriched in the phytohormone signaling and phenylpropanoid biosynthesis pathways. According to these results, GA treatment initially had an impact on carbon metabolism before having a major impact on phytohormone signaling and phenylpropanoid production (Appendix A).

Furthermore, DEG gene clustering and KOG analysis revealed eight different expression profiles. Among these, expression profiles 10, 19, 6, and 17 indicated an increasing trend, with signal transduction pathways and translational modifications, protein turnover, and chaperone proteins being particularly abundant. Expression profile 2 showed a declining trend, mainly enriched in RNA processing and modification and transcription, indicating that the cell division process was affected (Appendix A).

### 2.4. GA Regulates Cell Elongation in the Second Internode of N. cadamba

We found that GA promoted the growth of *N. cadamba* and the length of the second internode was increased (Figure 1B). To further analyze the growth of the second internode, we first examined the altered xylem anatomy induced by GA treatment. Although cross section analysis revealed no significant difference between the GA group and CK group at 7 d and 14 d in the xylem structures (Figure 3A), longitudinal section analysis revealed that the GA group had longer cell type at 7 d and 14 d compared to the CK group (Appendix A). The cell length from CK group was 75.23 ± 7.98 μm and 64.22 ± 2.96 μm at 7 d and 14 d, while the cell length from GA group was 89.57 ± 10.53 μm and 73.56 ± 3.71 μm at 7 d and 14 d, respectively (Figure 3B,C). These results indicated that GA promoted the second internode growth by increasing the cell length.

To further investigate the molecular mechanism of cell elongation in the second internode in *N. cadamba* plants treated with GA, the expression level of 10 genes related to cell elongation was analyzed. These genes were up-regulated and significantly changed and include *bHLH74* (*evm.TU.Contig555.524* and *evm.TU.Contig12.423*), *bHLH49* (*evm.TU.Contig54.415*, *evm.TU.Contig154.647*, and *evm.TU.Contig184.573*), *FAD_binding_4* (*evm.TU.Contig166.1*, *evm.TU.Contig969.62*, and *evm.TU.Contig161.87*) and *COBRA* (*evm.TU.Contig66.420* and *evm.TU.Contig52.64*) (Figure 3D, Appendix A). These results suggested that exogenous GA application regulates the expression of genes involved in cell elongation in *N. cadamba*.

### 2.5. DEGs Related to GA Biosynthesis and Signaling Transduction

From RNA-seq data, 45 genes associated with GA biosynthesis and 24 genes associated with signaling transduction were identified by analyzing DEGs expression patterns in *N. cadamba* (Appendix A). These DEGs were mainly expressed in GA3d vs. CK3d, GA7d vs. CK7d, and GA14d vs. CK14d groups.

By analyzing the expression level of DEGs from GA biosynthesis pathway with log2(Fold Change). The expression level of enzyme ENT-COPALYL DIPHOSPHATE SYNTHASE (CPS) (*evm.TU.Contig69.93*, *novel.2133*, *evm.TU.Contig421.376*, *evm.TU.Contig421.375*, and *evm.TU.Contig341.423*) gene was found to be up-regulated in CK1d vs. GA1d or down-regulated in CK14d vs. GA14d. The Enzyme ENT-KAURENE SYNTHASE (KS) enzyme related gene (*evm.TU.Contig117.8*, *evm.TU.Contig69.98_evm.TU.Contig69.99*, *evm.TU.Contig333.12*, and *evm.TU.Contig917.38*) was up-regulated in CK14d vs. GA14d. Likewise ENT-KAURENE OXIDASE (KO) (*evm.TU.Contig555.344*, *and evm.TU.Contig14.60*) showed differential expression. The enzyme ENT-KAURENOIC ACID OXIDASE (KAO) catalyzed GA precursor, *evm.TU.Contig184.687* and *evm.TU.Contig969.25,* were found to be up-regulated, while *evm.TU.Contig51.77*, *evm.TU.Contig797.34*, and *evm.TU.Contig51.78* were down-regulated in GA14d vs. CK14d. (Figure. 4) In addition, the expression of *GA20ox* (*evm.TU.Contig256.161*, *evm.TU.Contig96.467*, *evm.TU.Contig471.71*, and *evm.TU.Contig96.466*) was down-regulated, and *GA20ox* (*evm.TU.Contig477.208* and *evm.TU.Contig60.25*) was up-regulated in GA14d vs. CK14d. Furthermore, the expression of *GA3ox* (*evm.TU.Contig21.173* and *evm.TU.Contig46.95*) was down-regulated in GA7d vs. CK7d and GA14d vs. CK14d groups (Figure 4). Moreover, *GA2ox* acted as an inhibitor in the early steps of GA biosynthesis, depleting the substrates for bioactive GA and inactivated bioactive GAs [35]. From our RNA-seq data, *GA2ox* genes (*evm.TU.Contig208.88*, *evm.TU.Contig35.20*, *evm.TU.Contig271.154*, *evm.TU.Contig188.42*, and *evm.TU.Contig421.600*) were up-regulated in GA14d vs. CK14d, and (*evm.TU.Contig341.656*, *evm.TU.Contig96.100*, *evm.TU.Contig394.79*, and *evm.TU.Contig271.152*) were down-regulated (Figure 4). These results suggested that exogenous GA application significantly modified the expression of genes related to GA biosynthesis and inhibited endogenous GA biosynthesis.

By analyzing DEGs related to the GA signaling pathway expression level with log2(Fold Change), the GA receptor GID1 (*evm.TU.Contig421.435*, *evm.TU.Contig341.502*, and *evm.TU.Contig244.13*) was up-regulated in GA14d vs. CK14d (Appendix A). Notably, the GA repressor DELLA transcripts showed variable expression patterns as *evm.TU.Contig298.41*, *evm.TU.Contig553.11*, *evm.TU.Contig180.295*, *evm.TU.Contig259.48*, *evm.TU.Contig447.367*, *evm.TU.Contig55.380*, *evm.TU.Contig184.129*, and *evm.TU.Contig421.582* were down-regulated in GA14d vs. CK14d, while *evm.TU.Contig331.40* and *evm.TU.Contig16.807* were up-regulated in GA14d vs. CK14d (Figure 5, Appendix A). Above all, exogenous GA modulated the expression pattern of GA signaling regulators mainly by regulating GA receptor GID1 and repressor DELLA.

### 2.6. Exogenous GA Promotes Cell Division and Expansion

Cell division is closely related to plant growth regulation, and the cell cycle regulatory mechanism can play a direct role in plant morphogenesis and growth [36]. EXPANSINS act as cell wall expansion proteins and play role in various aspects of plant growth and development, and their expression is regulated by hormones [37,38]. EXPANSINS genes mostly regulate the elongation of the internode length in rice, with *OsEXPA4* being preferentially expressed in the elongation zone of rice internodes [38,39,40]. 

The cell division (Appendix A) and expansion (Appendix A) associated DEGs were analyzed to further investigate the mechanism by which exogenous GA enhances cell elongation of the second internode in *N. cadamba*. Interestingly, most of the DEGs associated with cell division, *CDC5*, *CDC6*, *CDC16*, *CDC20*, *CDC23*, *CDC45* and *CDC48* were up-regulated in the GA group, especially in GA7d vs. CK7d and GA14d vs. CK14d groups (Figure 6A). Furthermore, genes translated to EXPANSINS proteins were significantly up-regulated in GA1d vs. CK1d, GA7d vs. CK7d, and GA14d vs. CK14d groups (Figure 6B). These results suggested that exogenous GA regulated the expression of genes related to cell division and cell expansion to promote cell elongation of the second internode in GA1d vs. CK1d, GA7d vs. CK7d and GA14d vs. CK14d groups in *N. cadamba*.

### 2.7. DEGs with Phytohormone-Related Genes

Phytohormones often cross-talk to regulate a certain biological process. To examine the effects of GA treatment on *N. cadamba* and crosstalk with other phytohormones, we identified genes associated with phytohormone signaling and found the expression of 349 DEGs involved in auxin, cytokinin, GA, ABA, ethylene, JA, SA, and BR synthesis and signaling pathways were affected by exogenous GA treatment (Appendix A). Among them, auxin plays a crucial role in cell division and cell expansion [41,42], and we identified 104 auxin-related DEGs that were significantly up-regulated in response to the exogenous application of GA (Appendix A). Furthermore, DEGs involved in ethylene signaling, such as *ETR*, *EIN,* and *EBF*, were up-regulated, and most ERF transcription factors were down-regulated in GA14d vs. CK14d (Figure 7A). Nineteen DEGs related to JA signaling, *JAR1 JAZ,* and *MYC2* were significantly down-regulated in GA14d vs. CK14d (Figure 7B). 

The expression of closely associated genes varied during various cell differentiation phases, but the expression of 11 BR-related DEGs, *evm.TU.Contig477.674* and *evm.TU.Contig66.1083,* were up-regulated, while *evm.TU.Contig54.286*, *evm.TU.Contig12.510*, *evm.TU.Contig600.109*, *evm.TU.Contig600.107*, and *evm.TU.Contig600.108* were down-regulated (Appendix A). In addition, *BRI1 kinase inhibitor 1* (*BKI1*) (*evm.TU.Contig555.392* and *evm.TU.Contig14.98*) was up-regulated. Fourteen SA-related genes, such as *TGAL7* (*evm.TU.Contig44.177*), *TGA1* (*evm.TU.Contig12.185*), *TGA2.2* (*evm.TU.Contig577.15* and *evm.TU.Contig39.234*), and *TGAL4* (*evm.TU.Contig67.30*), were down-regulated (Appendix A). In conclusion, exogenous GA may affect other hormone signaling pathways to regulate morphology and development in *N. cadamba*.

### 2.8. Assay of DEGs Expression Level by qRT-PCR

Exogenous GA affects internode development to promote cell elongation, as revealed by RNA-seq data analysis of certain important DEGs. To further analyze the DEGs from the enriched pathway, we used qRT-PCT analysis of the transcript expression level by (Appendix A). Regarding the DEGs from the GA biosynthesis pathway, the KAO2 edited gene *evm. TU.Contig184.687* was up-regulated in all GA groups compared to CK groups; the GA20ox gene *evm.TU.Contig96.466* was down-regulated in GA groups at 3 d, 7 d, and 14 d compared to CK groups; and the expression of the CPS related gene *evm.TU.Contig421.375* had variable patterns in different GA groups compared to CK groups (Figure 8A). In addition, DEGs from GA signaling pathway (*evm.TU.Contig180.295*, *evm.TU.Contig553.11*, and *evm.TU.Contig331.40*) and phenylalanine biosynthesis pathway (*evm.TU.Contig210.8*, *evm.TU.Contig421.511*, and *evm.TU.Contig421.510*) were up-regulated in all GA groups compared to CK groups (Figure 8B,C). Likewise, the expression of DEGs involved in cell division and cell expansion (*evm.TU.Contig383.196*, *evm.TU.Contig66.420*, and *evm.TU.Contig154.647*) were up-regulated from all GA groups compared with CK groups (Figure 8D). Phytohormone-related DEGs, particularly those from the auxin signaling pathway, such as GH3.17 (*evm.TU.Contig101.17*) and SAURs (*evm.TU.Contig477.432 and evm.TU.Contig477.431*), were found to be up-regulated in all four GA groups when compared to CK groups (Figure 8E). These results revealed that exogenous GA plays an important role in regulating the second internode growth and development of *N. cadamba*, not only by modulating GA biosynthesis and signaling pathways but also through other processes such as phenylalanine biosynthesis, cell division, cell expansion, and phytohormone-related aspects. 

### 2.9. Changes in Related Transcription Factors and Candidate Genes 

Transcription factors (TFs) perform critical roles in plant growth and development by binding to the promoter domains of target genes and influencing transcription processes by increasing or inhibiting expression levels [43]. To confirm the changes in TFs after GA treatment, 1202 TFs were identified as differentially expressed in GA-treated plants (Appendix A). The TFs included 200 bHLHs, 131 ERFs, 83 bZIPs, 80 WRKYs, and 40 MYBs families. Notably, the expression of some bHLH, WRKY and MYB family genes was up-regulated (Figure 9). In addition, we selected 30 potential TFs with log2(|Fold Change|) ≥ 3 (Table 1), along with *TCP8* (*evm.TU.Contig63.469),* which has been described in the literature to participate in DELLA interactions [44]. Moreover, no interaction between the other 29 genes (bHLH, MYB, bZIP, AP2, and WRKY families) and GA has been identified. These might be the potential genes related to GA biosynthesis or signaling pathways for future investigation in *N. cadamba*.

## 3. Discussion

In this study, we report that exogenous GA regulates the internodal growth in *N. cadamba*. In the GA 7 d and GA 14 d groups, cell division and cell expansion are increased, resulting in increased second internode length. RNA-seq data analysis reveals that exogenous GA affects the endogenous GA biosynthesis and signaling pathway by regulating the expression of genes related to these pathways. Furthermore, we also identified and analyzed the transcriptional expression levels of genes associated with cell division and cell expansion. The signaling pathways of other phytohormones were also affected by alteration in the expression of genes involved in their synthesis and signaling pathways.

GA, as an important plant growth regulator, influences the growth pattern of plants at different developmental stages. Previous studies have proved that exogenous GA treatment regulates shoot organ development, promotes the activity of both subapical meristem and apical meristems, and thereby increases the number of stem units and bud length during vegetative bud growth in *Picea glauca* and *Pinus sylvestris* [45]. Moreover, exogenous GA_3_ stimulated the mitotic activity of the apical region and increased the rate and duration of cataphyll, resulting in wider and higher apical meristem formation [46]. However, exogenous GAs activated cell expansion and cell division in the cambial region, while they were not involved in xylem differentiation [47]. The application of GAs to plants under short daylight conditions leads to rapid stem elongation and flower formation [48]. Treatment of kale stem tips with GA_1_ and GA_4_ resulted in increased stem elongation [49]. However, the clear mechanism of the effect of exogenous GA application on the growth of *N. Cadamba* is still unclear. In the present study, the effects of exogenous GA application on the growth of *N. Cadamba* were analyzed in depth at the physiological and molecular levels.

We identified 69 DEGs in the GA biosynthesis and signal transduction pathway, implying that exogenous GA treatment leads to changes in endogenous GA biosynthesis and signal transduction. Increasing evidence suggests that GA biosynthesis is negatively regulated by GA activity. This feedback regulation was first discovered in a class of GA-insensitive semi-dominant dwarf mutants, for example, D8 in maize, GAI in *Arabidopsis,* and Rht in wheat [50,51]. These mutants have a phenotype comparable to GA-deficient dwarfism but have abnormally high amounts of bioactive GA, implying that decreased GA response leads to increased GA production and that GA activity lowers GA biosynthesis. The concentration of bioactive GA in GA-responsive tissues is tightly regulated through biosynthesis, transport, and inactivation. The mechanisms of regulation of GA biosynthesis and inactivation gene expression in higher plants in response to developmental and environmental signals is an active area of research. This process is largely outlined as the first step catalyzed by CPS and is thought to be the gateway to the GA biosynthetic pathway, playing a role in developmental regulation [52], but the synthesis of bioactive GA is limited by enzymes later in the pathway, particularly GA20ox [53]. In rice, mutations in the small granules of leucine zipper (HD-ZIP II) transcription factor and dwarf stalks cause dwarfism, which is associated with increased expression of several GA2ox genes and decreased expression of OsGA20ox1 and OsGA20ox2, although it is unclear whether this regulation is direct [54]. Peter Hedden provided a mechanism for GA internal environment stabilization through transcriptional regulation of GA metabolic genes by the GA signaling pathway: some GA3ox and GA20ox gene family members are down-regulated by GA signaling, while GA2ox genes are up-regulated [55,56]. Meanwhile, GA signaling promoted the degradation of the DELLA transcriptional regulator, which together with the transcription factor GAF1 up-regulated the expression of genes encoding GA3ox and GA20ox as well as the GA receptor GID1. We found that most GA20ox and GA3ox expressions were significantly down-regulated on days 7 and 14 of GA treatment during GA biosynthesis (Figure 4). This suggests that exogenous application of GA inhibited the expression of GA20ox and GA3ox, thereby suppressing the synthesis of endogenous active GA. Additionally, in the GA signaling pathway, we found that GID1-related gene expression was down-regulated, while DELLA-related gene expression was up-regulated, indicating that exogenous application of GA had a negative effect on GA signaling (Figure 5).

DELLA binds to light signaling regulator PIFs and BR signaling regulator BZR1 promoters as a repressor in the GAs signaling pathway, and the GA, BR, and light signaling integrated module, DELLA-BZR1-PIF4, inhibited target gene expression and regulated hypocotyl elongation. In the presence of GA, DELLA proteins were degraded, and PIFs and BZR1 were released from the DELLA negative interaction [57,58,59,60,61,62]. GAs and auxin work together to regulate vascular tissue formation during secondary cell wall expansion in plants [63]. Application of GAs promoted auxin polar transport to accumulate auxin in poplar stems [64]. The module of DELLA protein REPRESSOR of *ga1–3* Like 1 (RGL1), AUXIN RESPONSE FACTOR 7 (ARF7), INDOLE−3-ACETIC ACID 9 (IAA9), and RGL1-ARF7-IAA9 are involved in auxin and GA signaling and mediate cambial activity and development in poplar [65]. DELLA proteins directly interact with ethylene transcription factor ETHYLENE INSENSITIVE (EIN3) to inhibit the expression of *EIN3* targeted gene *HOOKLESS 1* (*HLS1*) and regulate apical hook development in *Arabidopsis* [66,67]. Furthermore, as an activator in the GAs signaling pathway, DELLA interacts with ABA signaling regulators *ABI3* and *ABI5* to activate expression of the *SOMNUS* gene and promote germination under stress conditions [12,68].

Our data showed that after the GA treatment (in 7d and 14d group), there was a significant increase in height and internodes elongation of *N. Cadamba* (Figure 1A–C). The shoot dry weight and shoot fresh weight were also significantly increased compared to CK (Figure 1D), which indicated that exogenous application of GA resulted in increased biomass of *N. Cadamba*. We also found that dry sections of *N. Cadamba* plants treated with GA for 7 d and 14 d showed no significant difference in lignification of the layer forming cells by histochemical staining, while cell length was increased significantly relative to CK (Figure 3B). GA stimulates cell elongation throughout the life cycle; this role is especially important because plant morphology is totally determined by cell division and elongation in the absence of cell mobility [69]. The promotion of stem growth by GA is based on the increase in cell length and the notion that GA is a regulator of cell elongation has been confirmed in most plants [70]. bHLH74 is a target gene of the miR396 family and a member of the basic Helix-Loop-Helix Transcription Factor family, which affects cell elongation and growth by directly regulating downstream genes, mainly through related hormone signaling pathways [71,72]. Other members of the bHLH family are also involved in the regulation of cell elongation growth, such as bHLH49. Our transcriptomic data showed that the expression of bHLH74 and bHLH49-related genes were significantly up-regulated in *N. Cadamba* after short-term GA treatment (7 d, 14 d) (Figure 3D). Notably, this is consistent with the results of histochemical staining of dry sections showing that GA treatment resulted in the elongation of cells in the second internode of *N. Cadamba* plants.

GA acts as a regulator of the cell division and expansion processes to enhance fruit size by determining the fruit cell number and size [73]. Exogenous GA is also involved in fruit enlargement through its ability to enhance cell expansion, which is similar to endogenous GA [74]. The fruit weight of Pyrus pyrifolia increased following GA treatment due to increases in cell size without cell number changes [75]. To accommodate cell expansion and to induce plant growth, expansins promote cells to overcome the constraints imposed by the rigid cell walls [76]. Here, in our study, expansins-related genes were significantly up-regulated and revealed the application of exogenous GA accelerated cell expansion (Figure 6B). In addition, in cucumber and tomato, GA production in the cotyledon plays a vital role in cell division during tissue reunion in the cortex [39]. In parthenocarpic citrus, the GA synthesis in the ovary walls at anthesis triggers cell division [77]. Our data show that genes of cell division were up-regulated also under the GA treatment (Figure 6A).

GA interacts with several other phytohormones and shows downstream or overlapping effects with auxin in many developmental responses [78,79]. The induction of cell elongation is one of the ultimate effects of GA, and to achieve these results, the cooperation of other hormones such as auxin may be a potential requirement. A recent study showed that DELLA and auxin proteins promote the accumulation of bioactive GA and independently regulate the GA synthesis pathway [80]. In addition, there are positive or negative interactions between GA and ethylene under different conditions [78]. Research analyses have shown that the application of exogenous ethylene to plants suppresses the expression of genes involved in GA metabolism, while GA treatment up-regulates some ethylene synthesis genes [78]. Consistent with this, we found that auxin-related genes were mostly up-regulated and interacted with GA to promote cell elongation, while ethylene-related genes were mostly up-regulated.

Based on our results, we propose a possible model for the physiological and molecular mechanisms of *N. cadamba* response to GA (Figure 10) to provide a foundation to further study the physiological consequences of *N. cadamba* growth of GA treatment.

## 4. Materials and Methods

### 4.1. Plant Material and Growth Conditions 

The *Neolamarckia cadamba* (*N. cadamba*) plant material was obtained from South China Agricultural University (Guangzhou, China). Plant bud tissues were cultured to seedings in rooting medium (MS basal medium (M519, Phytotech, KS, USA) 3% sucrose, 0.01% inositol, 0.8% agar and 0.1 mg/L NAA) for 15 d in a tissue culture room with the light of 5000 luxe, and the seedings grown outdoors for 7 d to acclimatize a plant to the growth chamber. Then, the plants were removed from the medium and cleaned in the liquid medium, fixed with a sponge and foam board in a hydroponic box, and cultured with pure water in a growth chamber with 25 °C, 16 h-light (30,000 lux light intensity), 8 h-dark and 70% atmospheric humidity condition in the greenhouse for 7 days [33]. Plants were cultured with Hoagland liquid nutrient solution. Additionally, the compositions of the liquid nutrient solution (4 mM Ca(NO_3_)_2_·4H_2_O, 6 mM KNO_3_, 2 mM MgSO_4_·7H_2_O, 95 µM MnSO_4_·4H_2_O, 1 mM NH_4_H_2_PO_4_, 80 µM NaFe-EDTA, 46.3 µM H_3_BO_3_, 0.8 µM ZnSO_4_·7H_2_O, 0.02 µM (NH_4_)6Mo_7_O_24_·4H_2_O and 0.3 µM CuSO_4_) were mixed and pH adjusted to 5.8.

### 4.2. GA Treatment

According to the findings of exogenous gibberellin on the growth and living process of strawberry, when the concentration of GA reaches 50 mg/L, the plants produce runners [81]. We chose to use a GA concentration of 50 mg/L for our experiments. Unified growth plants were selected and cultured in hydroponic boxes with sponges fixed. Plants were divided into two groups, the first group with 15 plants was cultured in liquid nutrient solution with 50 mg/L GA for treatment and named as the GA group, and another group with 15 plants was grown in the basal liquid nutrient solution used as a control and named as the CK group. Moreover, the liquid nutrient solution was continuously aired using an oxygen pump to ensure the plants obtained enough oxygen to grow. Additionally, the liquid nutrient solution was changed every three days. 

### 4.3. Plant Physiological Parameters Measurement

Plant samples from the CK group and GA group were collected after 7 d and 14 d cultured. Plant height and the second internode (counted from the top stem) length were observed and measured with a ruler. The shoot part from different samples was measured as fresh weight and dried under 65 °C condition for 3 d to measure as dry weight. The fresh/dry weight increasing rate is calculated with the formula ((7 d fresh/dry weight–0 d fresh/dry weight)/0 d fresh/dry weight, or (14 d fresh/dry weight–7 d fresh/dry weight)/7 d fresh/dry weight)). Each group samples from different cultured stages were conducted with three biological repeats at least.

### 4.4. Microscope Observation

The second internodes from the GA group and CK group after 7 d and 14 d cultured plants were fixed with 3% agarose in Petri dishes (704001, NEST Biotechnology, Wuxi, China) and cut to 40 µm thickness samples using a vibrating microtome (Leica VT1000S, Wetzlar, Germany). Samples were placed on slides, stained with 0.02% toluidine blue solution for 15 s and washed with ddH_2_O, and then observed with a scanning imaging microscopy system (Wanbang Junyi M8, Beijing, China). The cell number and cell area were analyzed with Image J.

### 4.5. RNA-Seq Analysis

Plant samples from the second internode of 1, 3, 7, and 14 d cultured GA group and CK group seedings were collected in liquid nitrogen with three biological repeats. Total RNA samples were extracted using RNAprep Pure Assay Kit (TIANGEN, Beijing, China). Novogene Company (Beijing, China) performed the RNA sample quality and transcriptome analysis. RNA-seq and differential expression genes (DEGs) analysis, referencing the published DNA sequence of *N. cadamba* [52], were performed using the Edger package in R software version.3.18 (Novogene Company, Beijing, China). Significant values were adjusted using the Benjamini and Hochberg methods with a differential expression threshold for values of *p* < 0.05. Heat map analysis of DEGs from different pathways was performed with TBtools [82]. Finally, the RNA-seq data were submitted to http://bigd.big.ac.cn/gsa/ with submission number CRA007183 accessed on 31 August 2022.

### 4.6. Differential Expression Genes Pathway Function Analysis

Gene function was annotated based on EuKaryotic Orthologous Groups, Swiss-Prot, Genomes and Kyoto Encyclopedia of Genes databases. KOG analysis was applied to the Self-organizing maps (SOM) cluster. The cluster was obtained by the k-means method. Cluster Profiler R package of R software was used for enrichment analysis of DEGs by setting a significance value of *p* < 0.05. Furthermore, a similar package was used for the estimation of GO enrichment and KEGG pathways analysis.

### 4.7. Quantitative Real Time PCR Analysis 

Total RNAs of the different cultured stages from the GA group and CK group were used to perform quantitative real time PCR (qRT-PCR). Total RNAs were reverse-transcribed and qRT-PCR analyses were performed using RT Kit with gDNA remover (AT311, TransGen Biotech, Beijing, China). Additionally, the expression level of the target genes were detected with SYBR type qRT-PCR mix (Q511, Vazyme, Nanjing, China). qRT-PCR reaction performed with LightCycler480 (Roche Molecular Biochemicals, Mannheim, Germany). Primers (Appendix A) were designed using the *SAMDC* gene from *N.cadamba* and NCBI (https://www.ncbi.nlm.nih.gov/, accessed on 31 August 2022) as the reference control gene and all target genes qRT-PCR analyses were conducted with three technical and biological repeats.

## Figures and Tables

**Figure 1 ijms-23-11842-f001:**
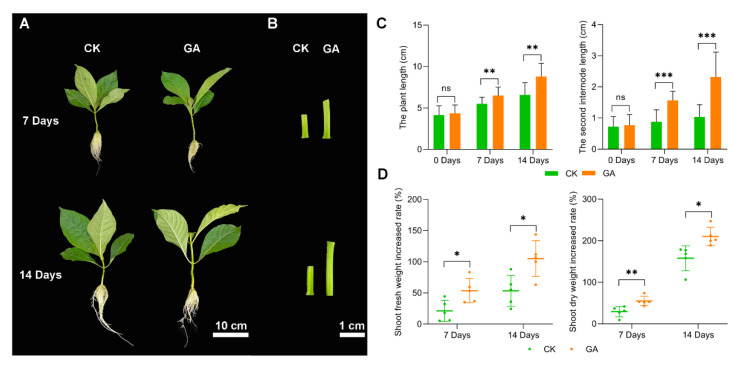
Growth of *N. cadamba* under exogenously applied 50 µM/L GA group (GA) and CK group (CK). (**A**) *N. cadamba* morphology of GA group and CK group at 7 d and 14 d. (**B**) The second internode of *N. cadamba* morphology of GA group and CK group at 7 d and 14 d. (**C**,**D**) Plant height, length of second internode section, shoot fresh weight increased rate, and shoot dry weight increased rate of CK group and GA group at different times. Results are mean ± SD of three biological replicates. Differences between mean values of CK and GA were compared using the student’s t-test (ns, *, **, and *** denote no significant difference, significant differences at *p* < 0.05, 0.01 and 0.001).

**Figure 2 ijms-23-11842-f002:**
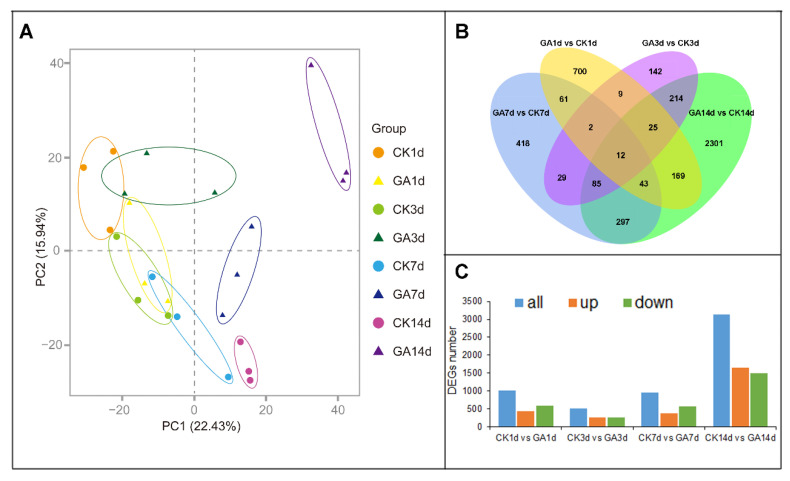
RNA-seq data analysis of the second internode from CK group and GA group in *N. cadamba*. (**A**) Principal component analysis (PCA) plots of RNA-seq data from *N. cadamba* second internode at 1 d, 3 d, 7 d and 14 d of CK group and GA group. (**B**) Venn diagram showing DEGs common or unique of the four comparative combinations (CK1d vs. GA1d, CK3d vs. GA3d, CK7d vs. GA7d, CK14d vs. GA14d). (**C**) DEGs analysis from RNA-seq data (both up- or down-regulated) for each comparative combination.

**Figure 3 ijms-23-11842-f003:**
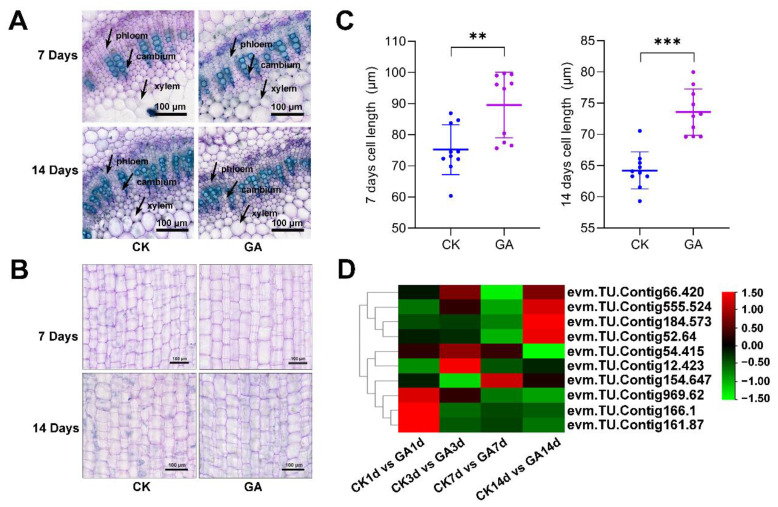
Effects of GA treatment on the second internode of *N. cadamba*. (**A**) Cross-section analysis of the second internode cell from CK group and GA group at 7 d and 14 d, material stained with toluidine blue. Arrows indicate phloem, cambium, and xylem. Scale bar, 100 μm. (**B**) Longitudinal section of the second internode cell from CK group and GA group at 7 d and 14 d. Material stained with toluidine blue. Scale bar, 100 μm. (**C**) The second internode cell length analysis from the CK group and GA groups at 7 d and 14 d. n = 15 and data are mean ± SD of three biological replicates as shown. Differences between mean values of CK and GA were compared using student’s *t*-test (** and *** denote significant differences at *p* < 0.01 and 0.001). (**D**) DEGs related to cell elongation were analyzed from RNA-seq data. Data are shown with log2 (Fold Change) analysis. Rows and columns of heatmap represent genes and samples, respectively.

**Figure 4 ijms-23-11842-f004:**
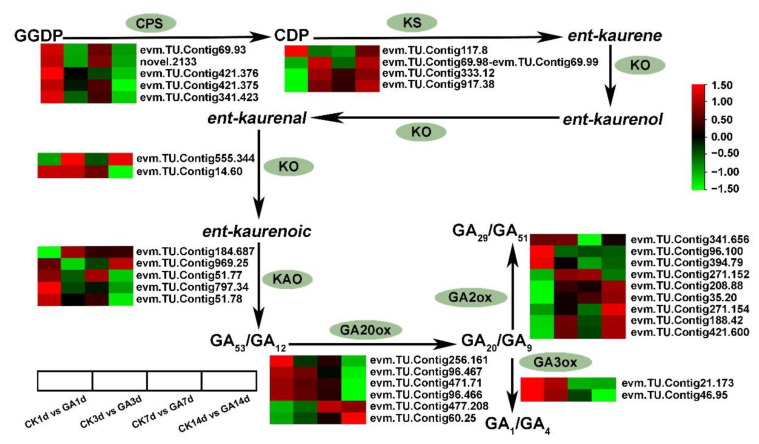
Simplified schematic diagram and DEGs heatmap analysis of GA biosynthesis. The GA biosynthesis components are marked with green background. Data is shown in log2 (Fold Change) analysis. Heatmap of rows and columns represent genes and samples, respectively. CPS, ENT-COPALYL DIPHOSPHATE SYNTHASE; KS, ENT-KAURENE SYNTHASE; KO, ENT-KAURENE OXIDASE; KAO, ENT-KAURENOIC ACID OXIDASE; GA20ox, GA−20 oxidase; GA2ox, GA−2 oxidase; GA3ox, GA−3 oxidase.

**Figure 5 ijms-23-11842-f005:**
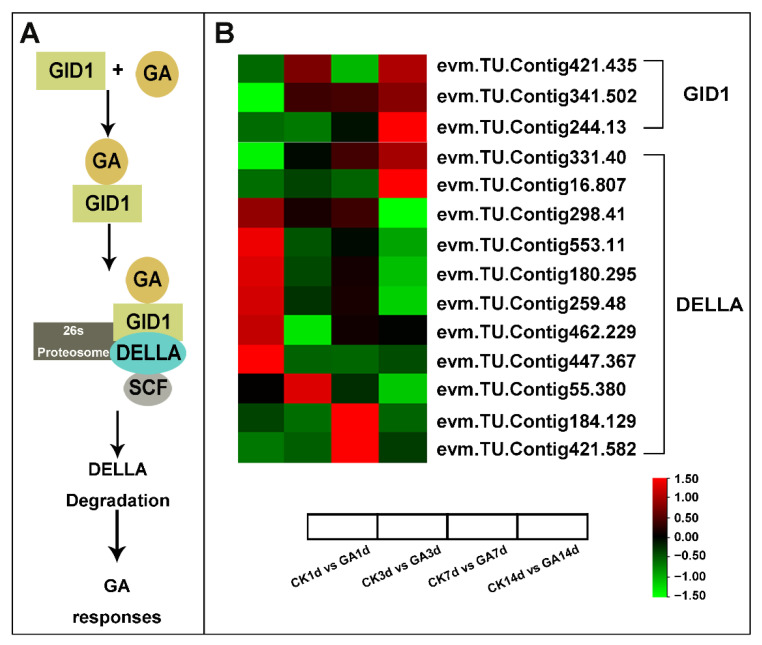
Simplified schematic diagram and DEGs heatmap analysis of GA signaling. (**A**) Simplified schematic diagram of the GA signaling pathway. (**B**) Heatmap analysis of GID1 and DELLA from RNA-seq data. Data is shown with log2 (Fold Change) analysis. Heatmap of columns and rows represent samples and genes, respectively.

**Figure 6 ijms-23-11842-f006:**
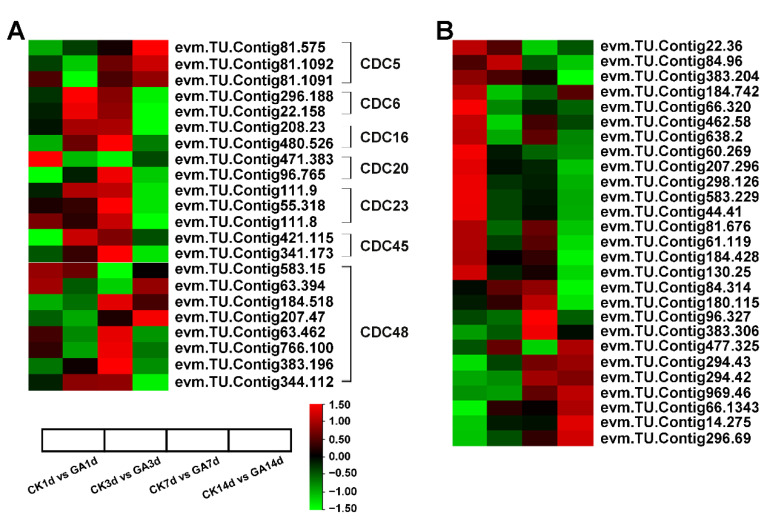
The analysis of DEGs associated with cell division and expansion on exogenous GA treatment from RNA-seq data. (**A**) The analysis of *CDCs* related to cell division on exogenous GA treatment. The expression of the mostly CDCs was up-regulated at GA7d vs. CK7d and GA14d vs. CK14d in *N. cadamba*. (**B**) The analysis of genes edited Expansins proteins involved in cell expansion on exogenous GA treatment. The expression of the mostly EXPANSINS genes was up-regulated at GA1d vs. CK1d, GA7d vs. CK7d, and GA14d vs. CK14d in *N. cadamba*. Data is shown with log2 (Fold Change) analysis. Heatmap of columns and rows represent samples and genes, respectively.

**Figure 7 ijms-23-11842-f007:**
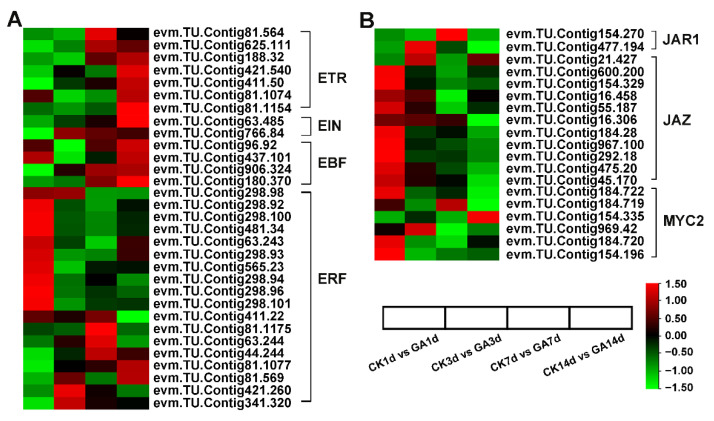
Effects of GA treatment on the other hormone signaling pathways. (**A**) RNA-seq data showed the DEGs of ethylene signaling. (**B**) RNA-seq data showed the DEGs of JA signaling. Data are shown with log2 (Fold Change) analysis. Heatmap of columns and rows represent samples and genes, respectively.

**Figure 8 ijms-23-11842-f008:**
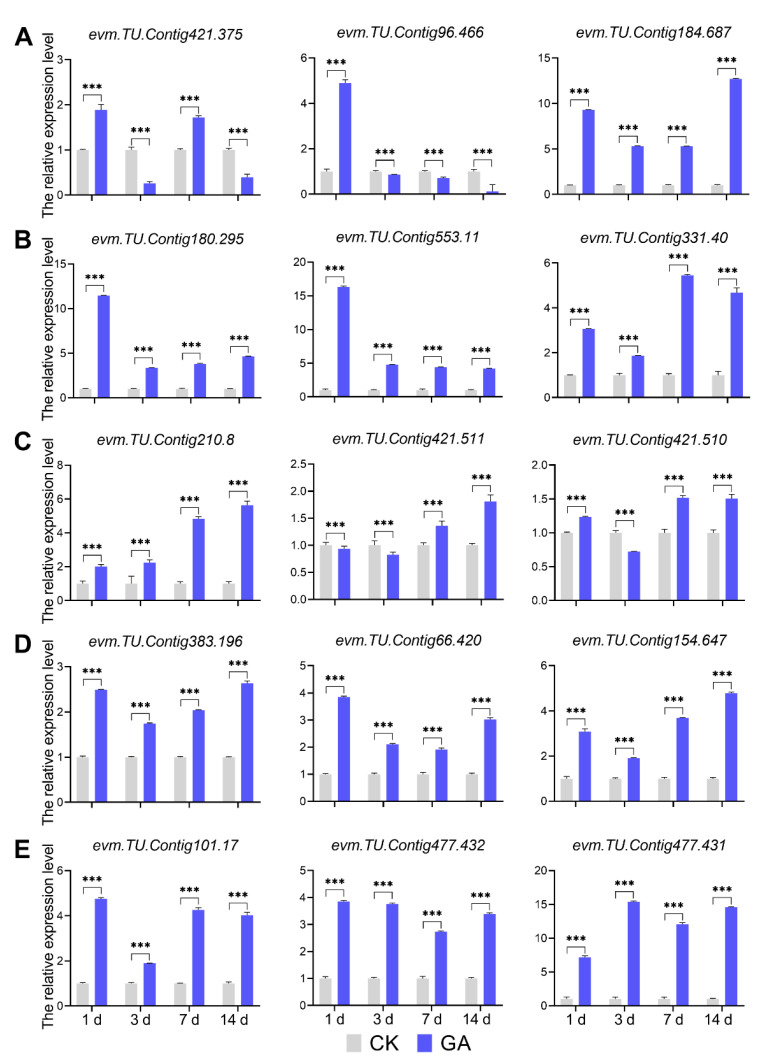
The analysis of DEGs from different pathways by qRT-PCR. (**A**) qRT-PCR-based the relative expression level analysis of genes involved in the GA biosynthesis pathway. (**B**) qRT-PCR-based the relative expression level analysis of genes involved in the GA signaling pathway. (**C**) qRT-PCR-based the relative expression level analysis of genes involved in the phenylalanine biosynthesis process. (**D**) qRT-PCR-based the relative expression level analysis of genes involved in cell division and cell expansion processes. (**E**) qRT-PCR-based the relative expression level analysis of genes involved in the auxin signaling process. CK, normal condition groups without GA treatment. GA, exogenous GA treatment groups applied 50 mg/L GA. 1d, GA1d vs. CK1d; 3d, GA3d vs. CK3d; 7d, GA7d vs. CK7d; 14d, GA14d vs. CK14d. Data are shown with the mean ± SD of three biological replicates. The significant differences between CK and GA were compared using student’s t-test (***, *p* < 0.001).

**Figure 9 ijms-23-11842-f009:**
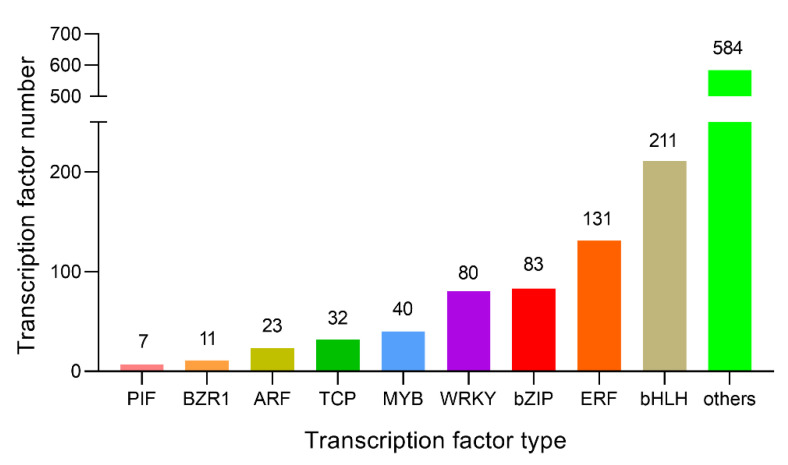
Statistics analysis of differentially expressed transcription factors. From the RNA-seq data analysis, 1202 differentially expressed TFs were significantly enriched, including PIF, BZR1, ARF, TCP, MYB, WRKY, bZIP, ERF, bHLH and other types of TF families.

**Figure 10 ijms-23-11842-f010:**
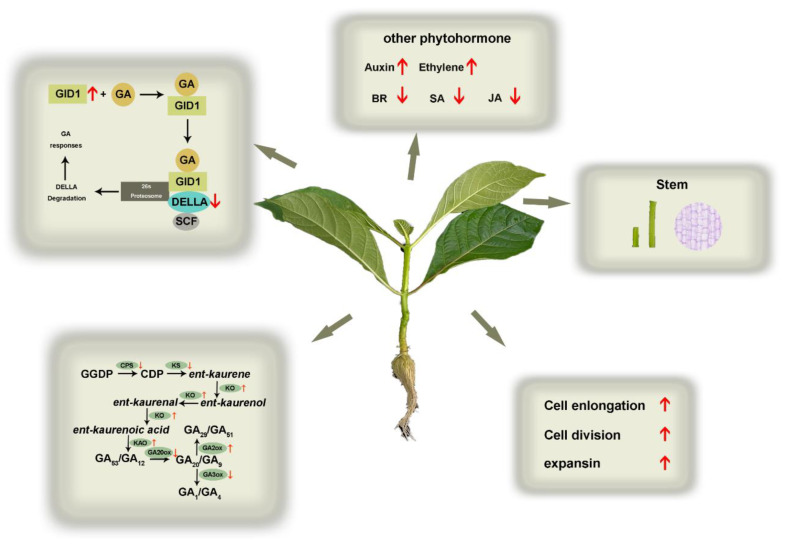
A hypothetical model of the effects of GA treatment on the physiological and molecular mechanisms of *N. cadamba*. Phenotypes and transcriptome changes in the second internode of GA treatment are indicated by the red arrows. Red arrows indicate an increase or a decrease in physiological indicators or gene expression.

**Table 1 ijms-23-11842-t001:** Thirty transcription factors expression analysis with log2 |Fold Change| ≥3 based on RNA-seq data.

Gene ID	GA1d vs. CK1d	GA3d vs. CK3d	GA7d vs. CK7d	GA14d vs. CK14d	Seq Description	TF Family
*evm.TU.Contig63.469*	2.46	0.64	−1.78	−3.27	TCP family transcription factor	TCP
*evm.TU.Contig84.232*	−2.16	−2.26	−3.17	−3.74	bHLH transcription factor bHLH051	bHLH
*evm.TU.Contig28.410*	−2.57	3.21	2.49	3.69	bHLH transcription factor bHLH025	bHLH
*evm.TU.Contig96.268*	−2.47	−2.40	−2.24	−3.87	bHLH transcription factor bHLH051	bHLH
*evm.TU.Contig28.419*	−0.32	3.45	0.50	−0.39	bHLH transcription factor bHLH025	bHLH
*evm.TU.Contig143.21*	−1.97	0.44	1.97	6.37	Transcription factor MYB44	MYB
*evm.TU.Contig294.168*	2.52	2.44	−1.83	4.61	Transcription factor MYB86	MYB
*evm.TU.Contig171.196*	2.48	1.85	−3.43	4.71	Transcription factor MYB44	MYB
*evm.TU.Contig3.39*	0.20	0.90	−4.36	−0.93	MYB-related protein MYB4	MYB
*evm.TU.Contig279.118*	−1.63	1.18	−3.66	4.37	Transcription factor MYB44	MYB
*evm.TU.Contig12.430*	2.02	2.71	2.30	4.21	bZIP transcription factor 53	bZIP
*evm.TU.Contig44.177*	0.02	−2.77	−3.79	−2.52	bZIP transcription factor 70	bZIP
*evm.TU.Contig906.67*	0.45	1.74	1.17	3.32	bZIP transcription factor 39	bZIP
*evm.TU.Contig66.454*	−0.99	−1.96	−1.80	−6.13	bZIP transcription factor 43	bZIP
*evm.TU.Contig81.569*	−3.72	−0.45	−1.27	−4.38	Ethylene-responsive transcription factor 1 B	ERF
*evm.TU.Contig66.1116*	1.02	−0.50	−4.03	0.14	Ethylene-responsive transcription factor 109	ERF
*evm.TU.Contig298.98*	0.66	0.72	4.19	4.08	Ethylene-responsive transcription factor 1 B	ERF
*evm.TU.Contig81.1174*	0.02	2.95	−2.84	−4.41	Ethylene-responsive transcription factor 098	ERF
*evm.TU.Contig383.343*	−0.45	−0.70	−3.56	−3.63	Ethylene-responsive transcription factor 1 A	ERF
*evm.TU.Contig298.96*	0.72	0.25	0.10	3.52	Ethylene-responsive transcription factor 1 B	ERF
*evm.TU.Contig600.195*	−1.61	0.11	−2.50	−3.49	Ethylene-responsive transcription factor 017	ERF
*evm.TU.Contig45.300*	−0.73	0.45	−0.71	−3.38	Ethylene-responsive transcription factor003	ERF
*evm.TU.Contig298.93*	−3.10	0.29	−3.44	3.10	Ethylene-responsive transcription factor 1 B	ERF
*evm.TU.Contig44.244*	−0.11	−1.08	−1.80	−3.17	Ethylene-responsive transcription factor 1 B	ERF
*evm.TU.Contig63.244*	2.36	−3.51	−0.50	−3.01	Ethylene-responsive transcription factor 1 B	ERF
*evm.TU.Contig298.94*	1.16	−0.21	0.03	3.60	Ethylene-responsive transcription factor 1 B	ERF
*evm.TU.Contig63.37*	0.02	−0.40	1.03	−3.20	Ethylene-responsive transcription factor 018	ERF
*evm.TU.Contig294.72*	2.46	4.76	−1.83	3.97	Ethylene-responsive transcription factor 2	ERF
*evm.TU.Contig435.24*	−0.68	0.15	−4.24	−3.64	WRKY transcription factor 72	WRKY
*evm.TU.Contig279.48*	0.57	2.46	−3.75	3.77	WRKY transcription factor 69	WRKY

## Data Availability

The data presented in this study are available on request from the corresponding authors.

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
