# Peer review of "Physiological and Transcriptomic Responses of Growth in *Neolamarckia cadamba* Stimulated by Exogenous Gibberellins"

_ijms, 2022, doi:10.3390/ijms231911842_

Round 1
Reviewer 1 Report
The manuscript by Lu Li et al. is a descriptive study summarizing the NGS experiment targeting gibberellin response genes in Kadamba tree (Neolamarckia cadamba). The manuscript has potential, and its premise of providing novel insights into gibberellin response genes in trees is of interest. Unfortunately, the present state of the manuscript is not acceptable for publication and should not be considered for a review without thorough proofreading and language editing.
Major issues:
1) The manuscript is hard to follow, the language is poor, and the presented content is missing points of interest. That is best reflected in the abstract, which should motivate the reader to access the main article and read it. In its present form, it does not provide reasonable information on why the study was conducted and what were the main results that provide evidence for differences in GA signaling pathways in this tree to those already described in the literature. That seems to be the main objective of this study, yet that is not reflected in the results or discussion. Some comparisons are missing - how many orthologous genes of GA signaling are present in this species compared to some of the best-studied models (e.g., Arabidopsis)? Is the GA response conserved or different? That should be included in the revised manuscript and discussed.
2) Figures and tables are not self-explanatory. For instance, Figure 3 should be replaced with a more meaningful output. It does not provide any easily accessible information (gene annotations are missing), and its 'main' objective of providing validation for NGS is not possible with the NGS data not being included. Many genes show oscillating regulation, and it is not clear to what extent these trends are reflected in the NGS dataset. Statistics should be included.
Figure 4D There are only eight sample groups in the dataset and all are included in the heatmap. What is being represented by the fold change? Missing: statistics, comprehensible gene annotation, description.
The legend for Table 1 is missing. Reported data are listed with nine! decimal places and without any statistical evaluation (standard deviation, t-test/ANOVA).
3) Materials and Methods
Employed statistics for analyzing time series are very simplistic and not suitable for NGS data.
Minor issues
- The reason for the selected GA concentration is not indicated or referenced.
- The nomenclature is incorrect (these are not seedlings).
- It is not clear how many plants per biological replicate were sampled.
- There are missing steps that should be described (e.g., data processing for PCA).
- Explain why were the plants treated with deionized water, as indicated in Materials and Methods (L404)
- There are more than 40 references in the introduction (a bit too much for a research article), yet the presented models are simplified and, at least to some extent, outdated (e.g., the GID-independent DELLA degradation pathway is missing).
- Only 20 out of 72 references represent a discussion of the results. That proportion is not optimal for a research article providing evidence of similarity/differences in well-known pathways.
In conclusion, this manuscript is not ready for publication. I recommend its complete revision, including statistical analyses and data comparison. I don't believe that this task would be achievable within the MDPI 'Major revision'.
Author Response
Reviewer 1
Comments and Suggestions for Authors
The manuscript by Lu Li et al. is a descriptive study summarizing the NGS experiment targeting gibberellin response genes in Kadamba tree (Neolamarckia cadamba). The manuscript has potential, and its premise of providing novel insights into gibberellin response genes in trees is of interest. Unfortunately, the present state of the manuscript is not acceptable for publication and should not be considered for a review without thorough proofreading and language editing.
Thanks for constructive suggestions. We improved the original manuscript in language editing and resubmitted the new manuscript to IJMS.
Major issues:
1) The manuscript is hard to follow, the language is poor, and the presented content is missing points of interest. That is best reflected in the abstract, which should motivate the reader to access the main article and read it. In its present form, it does not provide reasonable information on why the study was conducted and what were the main results that provide evidence for differences in GA signaling pathways in this tree to those already described in the literature. That seems to be the main objective of this study, yet that is not reflected in the results or discussion. Some comparisons are missing - how many orthologous genes of GA signaling are present in this species compared to some of the best-studied models (e.g., Arabidopsis)? Is the GA response conserved or different? That should be included in the revised manuscript and discussed.
Thanks for these constructive suggestions. We have corrected those mistakes and added more discussion in the manuscript.
2) Figures and tables are not self-explanatory. For instance, Figure 3 should be replaced with a more meaningful output. It does not provide any easily accessible information (gene annotations are missing), and its 'main' objective of providing validation for NGS is not possible with the NGS data not being included. Many genes show oscillating regulation, and it is not clear to what extent these trends are reflected in the NGS dataset. Statistics should be included.
Thanks for suggestions. We have corrected these mistakes.
Figure 4D There are only eight sample groups in the dataset and all are included in the heatmap. What is being represented by the fold change? Missing: statistics, comprehensible gene annotation, description.
Thanks for suggestions. We have re-analyzed the resulted with log2 (Fold Change) and added the statistical analysis in the manuscript.
The legend for Table 1 is missing. Reported data are listed with nine! decimal places and without any statistical evaluation (standard deviation, t-test/ANOVA).
Thanks for suggestions. We have added the legend for Table.1.
3) Materials and Methods
Employed statistics for analyzing time series are very simplistic and not suitable for NGS data.
Thanks for suggestions. We have corrected these mistakes.
Minor issues
- The reason for the selected GA concentration is not indicated or referenced.
Thanks for suggestions. We have corrected it.
- The nomenclature is incorrect (these are not seedlings).
Thanks for suggestions. We have corrected it.
- It is not clear how many plants per biological replicate were sampled.
Thanks for suggestions. We have corrected it. Each biological replicate with 15 plants.
- There are missing steps that should be described (e.g., data processing for PCA).
Thanks for suggestions. We analyzed the PCA using the Edger package in R software v.3.18. Significant values were adjusted using the Benjamini and Hochberg methods with a differential expression threshold for values of p< 0.05.and described it in the method section .
- Explain why were the plants treated with deionized water, as indicated in Materials and Methods (L404)
Thanks for suggestions. We have added the explanation in result 2.1 section.
- There are more than 40 references in the introduction (a bit too much for a research article), yet the presented models are simplified and, at least to some extent, outdated (e.g., the GID-independent DELLA degradation pathway is missing).
Thanks for these constructive suggestions. We have modified the manuscript.
- Only 20 out of 72 references represent a discussion of the results. That proportion is not optimal for a research article providing evidence of similarity/differences in well-known pathways.
Thanks for these constructive suggestions. We have modified the manuscript.
In conclusion, this manuscript is not ready for publication. I recommend its complete revision, including statistical analyses and data comparison. I don't believe that this task would be achievable within the MDPI 'Major revision'.
Many thanks for constructive suggestions. It is really very helpful for us to improve the paper.

Reviewer 2 Report
The submitted manuscript focused on the study of the influence and effect of GA in the key plant Neolamarckia cadamba. The study of the effect of growth regulators is still relevant, even though their discovery took place almost a century ago. As part of new, mainly genetic analyses, we are moving forward in our knowledge of the effect of growth regulators, while theoretical knowledge can be applied in practice. The manuscript will have a significant scientific contribution, but currently requires certain changes or additions. Given the fact that the issue of studying the influence of plant growth regulators is still in the forefront of interest, it is a shame that the authors cite rather older literature. I recommend checking the overview of the literature used and replacing some sources with newer ones. The graphical representation of the results is adequate, but it is a pity that some graphs, pictures and figures are small. I would supplement the results with some values. This applies, for example, to anatomical cuts. For them, it would be appropriate to supplement the description of the individual structures. The methodology is appropriate, but the text lacks a statistical analysis. The discussion is rather descriptive.
Author Response
Reviewer 2
Comments and Suggestions for Authors
The submitted manuscript focused on the study of the influence and effect of GA in the key plant Neolamarckia cadamba. The study of the effect of growth regulators is still relevant, even though their discovery took place almost a century ago. As part of new, mainly genetic analyses, we are moving forward in our knowledge of the effect of growth regulators, while theoretical knowledge can be applied in practice. The manuscript will have a significant scientific contribution, but currently requires certain changes or additions. Given the fact that the issue of studying the influence of plant growth regulators is still in the forefront of interest, it is a shame that the authors cite rather older literature. I recommend checking the overview of the literature used and replacing some sources with newer ones. The graphical representation of the results is adequate, but it is a pity that some graphs, pictures and figures are small. I would supplement the results with some values. This applies, for example, to anatomical cuts. For them, it would be appropriate to supplement the description of the individual structures. The methodology is appropriate, but the text lacks a statistical analysis. The discussion is rather descriptive.
Thanks for constructive suggestions. We have added more new literature to improve our manuscript. Besides, we represented the result with new graphs, pictures and figures, and redone the statistical analysis, such as the gene expression heatmap results were analyzed with log2(Fold Chang). In, addition to the discussion section have been modified with some deep description.
Reviewer 3 Report
The manuscript by Li et al. is devoted to an important problem: investigation of influence of exogenous gibberellins on plant growth including analysis of molecular basis of this influence. The work seems to be interesting; however, I have several comments and questions.
1. It is probable that background of using exogenous GA should be extended. Were there other works devoted to investigation of exogenous GA on plants? What results were earlier shown? What potential mechanisms of transport of exogenous GA into plant cells? Etc.
2. I do not sure that using “GA” in Title is optimal. Maybe, using “exogenous gibberellins” is more suitable.
3. Introduction: It is interesting: Can GA influence photosynthetic processes which are important role in growth? Considering relations between GA and JA (see Figure 10), indirect influence GA on photosynthesis seems to be possible. It should be discussed in Introduction.
4. P. 14, lines 402-405: Scheme of plant cultivation is not clear. What conditions were used for first 15 days of cultivation? Why plants were cultivated under outdoor conditions for next 7 days? Why the following cultivation was in the growth chamber? I suppose that scheme of plant cultivation should be described and explained in details.
5. Only 50 mg/L GA concentration seems to be used. Why this concentration was investigated? It should be clarified.
6. P. 15, lines 423-426: The equations “((7 d fresh/dry weight - 0 d fresh/dry weight)/0 d fresh/dry weight” and “(14 d fresh/dry weight - 7 d fresh/dry weight)/7 d fresh/dry weight))” is not clear. Maybe, it is error. It seems that authors did not investigate fresh/dry weight ratios. In accordance with Figure 1D authors rather investigated fresh weight and dry weight separately. If I am right then equations “100%*((7 d fresh weight - 0 d fresh weight)/0 d fresh weight” and “100%*(14 d fresh weight - 7 d fresh 425 weight)/7 d fresh weight))” (and analogical equations for dry weight) should be used.
Author Response
Reviewer 3
Comments and Suggestions for Authors
The manuscript by Li et al. is devoted to an important problem: investigation of influence of exogenous gibberellins on plant growth including analysis of molecular basis of this influence. The work seems to be interesting; however, I have several comments and questions.
Thanks for comments.
- It is probable that background of using exogenous GA should be extended. Were there other works devoted to investigation of exogenous GA on plants? What results were earlier shown? What potential mechanisms of transport of exogenous GA into plant cells? Etc.
Thanks for suggestions and this is our mistakes. We have cited more publications to explain the function of exogenous GA on plants in introduction and discussion sections.
- I do not sure that using “GA” in Title is optimal. Maybe, using “exogenous gibberellins” is more suitable.
Thanks for suggestions. We have corrected it.
- Introduction: It is interesting: Can GA influence photosynthetic processes which are important role in growth? Considering relations between GA and JA (see Figure 10), indirect influence GA on photosynthesis seems to be possible. It should be discussed in Introduction.
Thanks for your suggestions. We have cited publications associated with GA affect photosynthetic processes in the introduction section.
- P. 14, lines 402-405: Scheme of plant cultivation is not clear. What conditions were used for first 15 days of cultivation? Why plants were cultivated under outdoor conditions for next 7 days? Why the following cultivation was in the growth chamber? I suppose that scheme of plant cultivation should be described and explained in details.
Thanks for your suggestions. Plant cultivation conditions were added in the method section. Plants were cultivated in tissue culture room for 15 days. The acclimatization process of plants grown outdoor for 7 days is aimed to make sure that plant can quickly adapt to the growth chamber environmental conditions.
- Only 50 mg/L GA concentration seems to be used. Why this concentration was investigated? It should be clarified.
Thanks for this suggestion. In term of the GA concentration, we performed pilot experiment with different GA concentrations (0, 25, 50, 75, 100 mg/L) and the concentration of 50 mg/L GA is the suitable for plant growth.
- P. 15, lines 423-426: The equations “((7 d fresh/dry weight - 0 d fresh/dry weight)/0 d fresh/dry weight” and “(14 d fresh/dry weight - 7 d fresh/dry weight)/7 d fresh/dry weight))” is not clear. Maybe, it is error. It seems that authors did not investigate fresh/dry weight ratios. In accordance with Figure 1D authors rather investigated fresh weight and dry weight separately. If I am right then equations “100%*((7 d fresh weight - 0 d fresh weight)/0 d fresh weight” and “100%*(14 d fresh weight - 7 d fresh 425 weight)/7 d fresh weight))” (and analogical equations for dry weight) should be used.
Thanks for the constructive suggestions. We have corrected these results in our manuscript.
Reviewer 4 Report
1. Specify in the abstract the stage of the experimental plant used.
2. Authors should follow journal author guidelines for the preparation of MS and italicize the sci names throughout the MS. Such mistakes drive the attention towards finding typos and thus technicalities are left. I think these should be checked prior to sending them to the reviewers. It is a waste of time.
3. There is no information on the species investigated here. No prior work on growth has been cited in this or related species. No background information is given on what is known about the growth and its molecular mechanism in this or related species. Is this the only species that grow faster? Did u check the mechanisms in other species e.g., Mikania micrantha? it is called a mile-a-minute vine due to its fast growth mechanisms.
4. L110. Data not shown. Similar grammatical mistakes are widely present in this submission. Authors are advised to take care of such mistakes.
5. L104 AND ONWARDS. Please write the % or fold increase in the characteristics.
6. On what bases do the authors decide the use of GA conc.? Did they perform pilot experiments? Please explain the bases of using the specific GA conc.
7. L182 and onwards. The sentence seems incomplete and confusing.
8. L204. In English, if the sentence starts with a number, it should be in words and not digits.
9. L209. What do the authors mean by that “In GA signaling pathway, most enzyme expressions involved in the biosynthesis was down-regulated”. Firstly, the sentence needs grammatical corrections. Secondly, are there GA biosynthesis-related genes in the GA signaling pathway? The two are parts of two different pathways.
10. “parts of GA2ox genes () were activated….” Seems very non-scientific. Which parts of a gene are activated?
11. Where authors want to refer to multiple genes annotated as one enzyme/gene, they should preferably write them as transcripts annotated as genes e.g., We also found that three transcripts annotated as GID1 were highly expressed and downregulated…. You also need to correct this sentence from a grammatical point of view. What the authors want to write here is not understood. Highly expressed and downregulated are two contrasting and confusing terms. Authors must simply say that the GID1 transcripts were downregulated in GA-treated samples as compared to CK on day 3, day 7, and day 14. It should be very clear to readers in which the transcripts are downregulated; in CK or the treatments?
12. L222. Segregated is not the appropriate word. You can say the DELLA transcripts showed variable expression patterns. Also, the next sentence seems either incomplete or not well written.
13. When explaining the expression trends, authors should also indicate if the mean FPKM values (expressions) of a gene increased with the days or not and give a justification for the result.
14. What do authors mean by “part”? this is written frequently in the R section.
15. I think section 2.6. needs attention from the author. It should be revised. Authors should discuss the two pathways in separate paragraphs. First, they must show results on the expression trends of GA biosynthesis and conclude. Then, the same should be done for GA signaling in a new paragraph. And conclude both in an understandable way.
16. Figure 5 (also other figures with heatmaps). The heatmaps are based on the expressions/FPKM values, which to my understanding is not appropriate. Authors should provide the log2FC values for the three treatment comparisons. This is because authors are not considering the expression trends with respect to days (As evident from all R sections) but they are focusing on CK vs GA treatments. Secondly, there is no information in the legend about what they used to prepare heatmaps i.e., relative expression, FPKM, or what. From the values, it can be seen that the range is too small, which is not usual for FPKM. Anyway, it is recommended to use only three columns i.e., CK3vsGA3, CK7vsGA7, and CK14vsGA14 and use only Log2FC to prepare heatmaps. That will surely allow readers to see if expression increased in GA treatments as compared to CK or not.
17. What roles are being played by the CPS, KS, Kos, and KAO? Are they showing any reasonable trends that should be discussed?
18. Same comment as of 16 for Figure 6.
19. We already know GA responsive genes in plants, especially in Arabidopsis. Authors must consider screening their RNA seq results and highlight the key GA responsive genes that are being up/down-regulated in the GA treatments as compared to CK.
20. SECTION 2.7. The relaxation of the plant cell walls is coordinated by multiple players and not only are a result of changes in the expression of expansins. See https://elifesciences.org/articles/03031; Int. J. Mol. Sci. 2020, 21, 1743; doi:10.3390/ijms21051743. For the genes that are associated with overall cell wall expansion and loosening please see https://www.nature.com/articles/s41598-017-11495-4 and cite if relevant.
21. Section 2.8. Major interplay that helps plants increase cell division and cell expansion/elongation is with auxins. It is recommended to specifically present the Auxin biosynthesis and signaling pathway-related genes and their interplay with GA-related pathways. There are detailed studies on the role of auxins in the discussed processes. See https://www.ncbi.nlm.nih.gov/pmc/articles/PMC5979272/ and https://www.ncbi.nlm.nih.gov/pmc/articles/PMC2857164/
22. L292. Split the sentence into two to make it understandable.
23. L294-5. Wrong statement. WRKY can interplay with GA. Authors must search individually for the interaction of the mentioned TFs with the GA and/or Auxin-related pathways since both phytohormones interplay to give the studied responses. You can say that in terms of internode elongation, there are limited studies, but you can’t say that GA and these TFs have no interaction. Even if not studied in the case of internode, you must explain possible interaction and indicate its necessity to explore in future studies.
24. Overall, the MS needs significant rework from the authors. They have got useful results but the way they explained the results is sometimes not appropriate or confusing. All the heatmap figures should be based on Log2FC and not the other criteria. Language is weak and distracts the reader.
Author Response
Reviewer 4
Comments and Suggestions for Authors
- Specify in the abstract the stage of the experimental plant used.
Thanks for this suggestion and we have corrected this mistake.
- Authors should follow journal author guidelines for the preparation of MS and italicize the sci names throughout the MS. Such mistakes drive the attention towards finding typos and thus technicalities are left. I think these should be checked prior to sending them to the reviewers. It is a waste of time.
Thanks for the constructive suggestions. We have improved the manuscript. We are sorry for these mistakes from the manuscript become trouble for your reviewing.
- There is no information on the species investigated here. No prior work on growth has been cited in this or related species. No background information is given on what is known about the growth and its molecular mechanism in this or related species. Is this the only species that grow faster? Did u check the mechanisms in other species e.g., Mikania micrantha? it is called a mile-a-minute vine due to its fast growth mechanisms.
Thanks for the constructive suggestions. We have corrected these mistakes and cited more reference about this plant information.
- L110. Data not shown. Similar grammatical mistakes are widely present in this submission. Authors are advised to take care of such mistakes.
Thanks for this suggestion and we have corrected this mistake.
- L104 AND ONWARDS. Please write the % or fold increase in the characteristics.
Thanks for this suggestion and we have corrected this mistake.
- On what bases do the authors decide the use of GA conc.? Did they perform pilot experiments? Please explain the bases of using the specific GA conc.
Thanks for this suggestion. In term of the GA concentration, we performed pilot experiment with different GA concentrations (0, 25, 50, 75, 100 mg/L) and the concentration of 50 mg/L GA is the suitable for plant growth.
- L182 and onwards. The sentence seems incomplete and confusing.
Thanks for this suggestion and we have corrected this mistake.
- L204. In English, if the sentence starts with a number, it should be in words and not digits.
Thanks for this suggestion and we have corrected this mistake.
- L209. What do the authors mean by that “In GA signaling pathway, most enzyme expressions involved in the biosynthesis was down-regulated”. Firstly, the sentence needs grammatical corrections. Secondly, are there GA biosynthesis-related genes in the GA signaling pathway? The two are parts of two different pathways.
Thanks for the constructive suggestions. We have improved this section.
- “parts of GA2ox genes () were activated….” Seems very non-scientific. Which parts of a gene are activated?
Thanks for this suggestion and we have corrected this mistake.
- Where authors want to refer to multiple genes annotated as one enzyme/gene, they should preferably write them as transcripts annotated as genes e.g., We also found that three transcripts annotated as GID1 were highly expressed and downregulated…. You also need to correct this sentence from a grammatical point of view. What the authors want to write here is not understood. Highly expressed and downregulated are two contrasting and confusing terms. Authors must simply say that the GID1 transcripts were downregulated in GA-treated samples as compared to CK on day 3, day 7, and day 14. It should be very clear to readers in which the transcripts are downregulated; in CK or the treatments?
Thanks for the constructive suggestions and we have corrected these mistakes.
- L222. Segregated is not the appropriate word. You can say the DELLA transcripts showed variable expression patterns. Also, the next sentence seems either incomplete or not well written.
Thanks for this suggestion and we have corrected this mistake.
- When explaining the expression trends, authors should also indicate if the mean FPKM values (expressions) of a gene increased with the days or not and give a justification for the result.
Thanks for the constructive suggestions and we have corrected these mistakes.
- What do authors mean by “part”? this is written frequently in the R section.
Thanks for the constructive suggestions and we have corrected these mistakes.
- I think section 2.6. needs attention from the author. It should be revised. Authors should discuss the two pathways in separate paragraphs. First, they must show results on the expression trends of GA biosynthesis and conclude. Then, the same should be done for GA signaling in a new paragraph. And conclude both in an understandable way.
Thanks for the constructive suggestions. We have improved this section.
- Figure 5 (also other figures with heatmaps). The heatmaps are based on the expressions/FPKM values, which to my understanding is not appropriate. Authors should provide the log2FC values for the three treatment comparisons. This is because authors are not considering the expression trends with respect to days (As evident from all R sections) but they are focusing on CK vs GA treatments. Secondly, there is no information in the legend about what they used to prepare heatmaps i.e., relative expression, FPKM, or what. From the values, it can be seen that the range is too small, which is not usual for FPKM. Anyway, it is recommended to use only three columns i.e., CK3vsGA3, CK7vsGA7, and CK14vsGA14 and use only Log2FC to prepare heatmaps. That will surely allow readers to see if expression increased in GA treatments as compared to CK or not.
Thanks for the constructive suggestions and we have corrected these mistakes.
- What roles are being played by the CPS, KS, Kos, and KAO? Are they showing any reasonable trends that should be discussed?
Thanks for the constructive suggestions and we have cited these references related to CPS, KS, KOS and KAO in the introduction section and discussion in the result and discussion sections.
- Same comment as of 16 for Figure 6.
Thanks for the constructive suggestions and we have corrected these mistakes.
- We already know GA responsive genes in plants, especially in Arabidopsis. Authors must consider screening their RNA seq results and highlight the key GA responsive genes that are being up/down-regulated in the GA treatments as compared to CK.
Thanks for your comments. We have modified this manuscript and emphasize the important genes is discussion section.
- SECTION 2.7. The relaxation of the plant cell walls is coordinated by multiple players and not only are a result of changes in the expression of expansins. See https://elifesciences.org/articles/03031; Int. J. Mol. Sci. 2020, 21, 1743; doi:10.3390/ijms21051743. For the genes that are associated with overall cell wall expansion and loosening please see https://www.nature.com/articles/s41598-017-11495-4 and cite if relevant.
Thanks for the constructive suggestions and we have cited these references.
- Section 2.8. Major interplay that helps plants increase cell division and cell expansion/elongation is with auxins. It is recommended to specifically present the Auxin biosynthesis and signaling pathway-related genes and their interplay with GA-related pathways. There are detailed studies on the role of auxins in the discussed processes. See https://www.ncbi.nlm.nih.gov/pmc/articles/PMC5979272/ and https://www.ncbi.nlm.nih.gov/pmc/articles/PMC2857164/.
Thanks for the constructive suggestions and we have cited these references.
- L292. Split the sentence into two to make it understandable.
Thanks for the constructive suggestions and we have modified this part.
- L294-5. Wrong statement. WRKY can interplay with GA. Authors must search individually for the interaction of the mentioned TFs with the GA and/or Auxin-related pathways since both phytohormones interplay to give the studied responses. You can say that in terms of internode elongation, there are limited studies, but you can’t say that GA and these TFs have no interaction. Even if not studied in the case of internode, you must explain possible interaction and indicate its necessity to explore in future studies.
Thanks for your constructive suggestions. Indeed, we are prepared to explore these genes’ function in N. cadamba.
- Overall, the MS needs significant rework from the authors. They have got useful results but the way they explained the results is sometimes not appropriate or confusing. All the heatmap figures should be based on Log2FC and not the other criteria. Language is weak and distracts the reader.
Many thanks for constructive suggestions. It is really very helpful for us to improve the paper.
Round 2
Reviewer 1 Report
I have not found a version of the manuscript with tracked changes, but it seems that most changes were only cosmetic. The authors have not solved the major issue, and the manuscript still requires professional proofreading and editing to improve its legibility and correct errors in style and grammar.
Some issues were addressed by removing the presented data. That is questionable but probably acceptable for the descriptive study and the publication level. My concerns about statistics were addressed by improving the M&M section.
The abstract has not been improved and does not motivate a reader to read the manuscript. Its structure is strange (what is the purpose of numbers in brackets?).
Finally, the hypothetical model of GA action summarized in Figure 10 does not seem to be different from the general mode of GA response. I asked the authors to highlight the novelty, and that is still missing. Please, do that, at least for this part. Show the canonical GA response and indicate (by comparison) what is different in your model organism.
Author Response
I have not found a version of the manuscript with tracked changes, but it seems that most changes were only cosmetic. The authors have not solved the major issue, and the manuscript still requires professional proofreading and editing to improve its legibility and correct errors in style and grammar.
Many thanks for your constructive suggestions. It is very helpful for us to improve the quality of paper. We have corrected most of the style and grammar mistakes to a better version.
Some issues were addressed by removing the presented data. That is questionable but probably acceptable for the descriptive study and the publication level. My concerns about statistics were addressed by improving the M&M section.
Thanks for your comments. We removed some statement and corrected most of the results and improved the M&M section, because most of them were not comprehensively described.
The abstract has not been improved and does not motivate a reader to read the manuscript. Its structure is strange (what is the purpose of numbers in brackets?).
Thanks for your comments. The numbers in brackets were meet the requirements of journal submission.
Finally, the hypothetical model of GA action summarized in Figure 10 does not seem to be different from the general mode of GA response. I asked the authors to highlight the novelty, and that is still missing. Please, do that, at least for this part. Show the canonical GA response and indicate (by comparison) what is different in your model organism.
Thanks for constructive suggestions. Due to the GAs play a crucial role in plant growth and development, we concluded that the physiological and molecular mechanisms of exogenous GA treatment stimulation in N. cadamba.
Reviewer 4 Report
The figure quality is too bad. Especially the resolution of most of the figures is not up to the standard. Should be improved as per author guidelines.
Author Response
Thanks for your constructive suggestions. We replaced most of the figures in last version with high quality figures.
Round 3
Reviewer 1 Report
I am not satisfied with the modifications, and I don't believe that the manuscript has been significantly improved. The legibility of some parts of the manuscript is still very poor (e.g., L167-L224). The requested comparison with previous models of GA signaling has not been done, and the novelty or unique features of GA signaling in N. cadamba are not presented. However, I don't believe that my comments will have any impact on the manuscript, and I withdraw my request for revisions.